# A New Era for Mild Strain Cross-Protection

**DOI:** 10.3390/v11070670

**Published:** 2019-07-23

**Authors:** Katrin Pechinger, Kar Mun Chooi, Robin M. MacDiarmid, Scott J. Harper, Heiko Ziebell

**Affiliations:** 1The New Zealand Institute for Plant and Food Research Limited, Auckland 1142, New Zealand; 2School of Biological Sciences, University of Auckland, Auckland 1142, New Zealand; 3Department of Plant Pathology, Washington, State University, Prosser, WA 99350, USA; 4Julius Kühn Institute, Institute for Epidemiology and Pathogen Diagnostics, Messeweg 11-12, 38104 Braunschweig, Germany

**Keywords:** cross-protection, mild strain cross-protection, plant protection

## Abstract

Societal and environmental pressures demand high-quality and resilient cropping plants and plant-based foods grown with the use of low or no synthetic chemical inputs. Mild strain cross-protection (MSCP), the pre-immunization of a plant using a mild strain of a virus to protect against subsequent infection by a severe strain of the virus, fits with future-proofing of production systems. New examples of MSCP use have occurred recently. New technologies are converging to support the discovery and mechanism(s) of action of MSCP strains thereby accelerating the popularity of their use.

## 1. Societal and Environmental Pressures for Resilient Cropping Plants

Multiple and significant global trends are currently demanding for highly sustainable, reliable, and increasingly efficient production of plant crops. Our response to the specter of climate change drives the need for reduced greenhouse gas emissions e.g., in the production or use of synthetic chemicals for insect management, and the need to be able to grow crops under progressively extreme climatic conditions [1,2]. World population growth drives the need for increased food production on less land as more people require housing and infrastructure [3]. Adding to the demands on plant production, there is a move to healthier and more sustainable sources of protein from plants rather than meat, and plant crops are the main sources of increasing biofuel production [4]. Sustainable production methods that reduce the need for synthetic chemicals are desired to reduce habitat degradation and harm to people [5]. The world’s increasing middle class population can afford high-quality sustainably produced food, while the world’s poorest population require secure food supplies at lowest costs [6]. Sustainable production of plant-based food is required to care for the environment and for future generations of people [7].

While viruses are the major pathogen causing emerging infectious diseases in plants worldwide, we have a few methods to manage the crop viruses [8]. Furthermore, it is the world’s poorest people who depend on the staple crops that often suffer from devastating losses because of virus infections. Even in industrialized countries, plant viruses cannot be managed easily. To meet the United Nation’s Sustainable Development goal 2, Zero Hunger, we need to implement crop virus management methods [9]. We believe that now is the era for mild strain cross-protection (MSCP), which once established can provide “low tech” and sustainable protection against pathogenic viruses of crop plants.

Cross-protection, also known as “pre-immunization” is a method of plant virus control, whereby a plant is deliberately infected with a mild strain of a virus in order to protect the plant against damage caused by a more severe (“challenge”) strain of the same virus [10]. MSCP provides a sustainable pathway among many needed to address both the societal and ecological resilience that will “increase the crop production while reducing unsustainable uses of water, nutrients, and agricultural chemicals” [11,12].

When naturally occurring mild strains are used, MSCP is not burdened with the social acceptability issues of genetic modification or gene editing. However, the use of MSCP does need to retain social license for growers and consumers. Social license cannot be assumed [13]. As such, several requirements to ensure long-lasting protection using MSCP have been identified [14]. Importantly, experimental demonstration of the protection afforded by the mild strain, its durability and impact on the crop and other hosts of the virus, among other safeguards detailed previously, must be undertaken in a rigorous manner [14]. To our knowledge, no report has been published describing the “escape” of a mild cross-protecting virus strain into the “wild” and causing severe symptoms on non-target hosts. This risk appears to be very low in comparison to the reported cases where MSCP broke down after a while due to the arrival of “novel” more severe challenge strains, incomplete spread within the host plant, or the occurrence of new viral vectors [15,16,17,18,19,20,21]. Therefore, only close working relationships between the crop producers and scientists can ensure the identification of mild strains of viruses (and the severe strains they protect against) and the durable use of MSCP in crop production systems. Open communication including information for general and specialized audiences about the uses, risks, and control measures for MSCP maintains the confidence of producers, consumers, and the public about its environmentally friendly and people-safe use in crop production.

## 2. What Is Mild Strain Cross-Protection?

Cross-protection is usually only effective between closely related strains of the same virus, although there are cases where phylogenetically distinct viruses can protect against each other, for example citrus vein enation virus was reported to protect against citrus tristeza virus (CTV) [22]. Exploiting the fact that cross-protection only occurred between closely related viruses, cross-protection was used historically as a virus diagnosis and characterization tool before the serological and nucleic acid-based methods were available. Supporting the MSCP for closely related species, Folimonova demonstrated that cross-protection is only effective between CTV isolates belonging to the same strain [23].

The traditional approach for identification of isolates for cross-protection involves identifying individual plants that display only mild symptoms, or remain symptomless, in fields where other neighboring plants display severe symptoms correlated with viral infection. Next, virus(es) are isolated from these individual plants; it should be noted that one part, limb or side of a plant may display mild symptoms while the rest of the plant exhibits severe symptoms, potentially indicating different sub-populations distributed throughout the individual plant [24]. These viruses are first isolated, then used to inoculate the target host of choice that is subsequently challenged with known pathogenic isolates. This challenge may be deliberate, through simultaneous or staggered mechanical, graft, or vector inoculation, or the putative protective isolate may be exposed to natural infection under field conditions over an extended period of time.

When searching for potential cross-protective isolates, one should consider that an isolate may only “protect” one host cultivar, for example in citrus the CTV cross-protective isolate IAC protects Pear sweet orange but not Hamlin or Valencia [25]. Furthermore, protection may be limited to pathogenic isolates present in one geographic location; the IAC isolate has provided stable protection in Sao Paulo state, but not the neighboring state of Paraná in Brazil [26,27].

Another method for selecting attenuated strains is by passing a viral population that may be a composite of many strains through an alternative host. Sometimes the predominant strain within the population changes through this virus–host interaction. Reasons for this may be that the alternative host is resistant to some strains of the virus and prohibits the replication and/or movement of such strains while others are able to move through the plant more rapidly. This technique was used by Yeh and Cheng [28] who used *Cucumis metuliferus* as an alternative host to obtain mild strains of papaya ringspot virus-P from nitrous acid-induced mild strains HA5-1 and HA6-1.

Attenuated (mild) strains may also be selected through thermal treatment of infected plants; in this way a heat-attenuated mild strain of CTV was discovered [29]. Additionally, mutagenic agents or DNA recombination can be used to create mild strains in the laboratory.

High-throughput sequencing (HTS) is a promising tool for the discovery of potential cross-protective isolates. HTS can rapidly identify the genotypic composition of the virus or viruses present [30,31,32]; a necessary precondition for both like-for-like superinfection exclusion [23] and population-interaction cross-protective mechanism hypotheses. Moreover, HTS permits the comparison, at the sequence level, between the prospective mild protective and pathogenic challenge isolates. However, there are limitations with this approach. The data represent a “snapshot” in space and time, that is, the virus or viruses present in a single plant sample. This is not necessarily representative of the entire plant because of variable titer and distribution, virus tropism, and/or seasonal fluctuations [33]. Furthermore, sequence data alone cannot, without knowledge of the mechanism of cross-protection in the plant host in question, identify what a given isolate can protect against. Finally, and most critically, HTS and subsequent bioinformatic analyses cannot yet reveal the biological properties of an isolate, i.e., whether it is pathogenic, and/or in what hosts it is able to replicate [34].

## 3. Mild Strain Cross-Protection—Discovery through to Examples from Laboratory and Field Cross-Protection

Folimonova [16] laid out a “recipe” for the cross-protection by CTV that outlines how a protective mild strain can be found and isolated; their approach involves biological characterization of severe and mild isolates and suggestions for separating individual genotypes. While their approach is specific to CTV, their “recipe” can be applied to other plant viruses.

The cross-protection phenomenon was observed almost a century ago [35,36,37,38] and has since seen multiple examples from various virus/crop combinations, in depth reviewed by Ziebell and Carr [10]. The past decade has witnessed a resurgence of MSCP with efficacy demonstrated in the research environment as well as field-deployed protection (Table 1). Examples of mild strains are given below as case studies to illustrate the methods to discover and select for mild strains, to commercialize the mild strains, and to examine potential mechanisms of MSCP.

## 4. Commercialization of Mild Strain Cross-Protection: Case Study Pepino Mosaic Virus

Pepino mosaic virus (PepMV), genus *Potexvirus*, was discovered in the 1970s by Jones, Koenig, and Lesemann [100]. Originally found on pepino (*Solanum muricatum*) plants in Peru, extensive host range studies showed that not only numerous *Solanum* species and other members of the *Solanaceae* family can be infected by PepMV but also artificial infection (albeit not systemic infection) of *Tetragonia expansa*, a member of the *Aizoaceae* family and *Cucumis sativus*, a member of the *Cucurbitaceae* were possible (although these hosts may not play a major role under natural conditions) [90]. As a potexvirus, PepMV is easily transmitted mechanically but not by common virus vectors such as aphids.

However, for two decades, PepMV lived in the shadows until an outbreak was reported in greenhouse tomatoes in the Netherlands [101]; the first report of natural infection of tomatoes by PepMV. Since then, PepMV has been found in virtually all tomato-producing European countries; the ease of mechanical transmission during handling of tomato crops as well as the use of untreated tomato seeds and the global trade of this high-value crop may also have contributed to the spread of PepMV within Europe and worldwide within a few years; transmission by pollinating bumblebees or soil-borne fungal vectors as well as infested irrigation water may play additional routes of rapid spread of PepMV within greenhouse facilities [66,102,103,104,105,106,107,108]. The economic impact of PepMV is immense as fruit symptoms such as marbled, discolored, or open fruits have a lower market value or cannot be sold at all [105,109,110,111].

Although several attempts were underway to find sources of genetic resistance to PepMV within accessions of tomato and wild relatives, to the best of our knowledge, no PepMV-resistant tomato varieties are yet available [102,103]. In addition, PepMV comprises at least four different genetic groups; different isolates cause different symptoms depending on hosts and these might harbor strains that can overcome potential resistance [105,109,110,112,113,114,115,116,117,118,119,120,121,122]. The observation of naturally occurring mild PepMV strains with little or no impact on fruit symptoms and/or yield led to the proposal to use those mild strains for cross-protection purposes [105,123,124,125]. Moreover, attempts have been carried out to produce “artificial” mild strains for the use as cross-protecting agents [69,125].

In 2015, the European Food Safety Authority (EFSA) concluded that there was no consumer risk in using a natural mild strain, PepMV CH2 isolate 1906 (provided the applied product was free from microbial contaminants) for MSCP, but also identified knowledge gaps in terms of environmental risk assessment or ecotoxicology [126]. Nevertheless, the mild strain was commercialized and licensed in 14 European countries, protecting more than 4500 ha of tomatoes since its introduction [127]. A second mild strain PepMV product VX1 was assessed by EFSA in 2017; in this case, no major risks associated with the application of this mild strain were identified [128]. The commercial product “V10” includes two different mild strains of PepMV, VX1 and VC1, and suppresses symptoms induced by different PepMV strains [129]. A similar approach of combining more than one mild strain to protect tomato crops from a range of PepMV isolates was recently adopted by Agüero and colleagues [68] who tested the ability of two mild isolates of PepMV (Sp-13 and PS5) in various field trials.

It is encouraging to see that MSCP works well in these crops and can hopefully also be applied to other viruses that pose threats to tomato production including the newly emerging tomato brown rugose fruit virus, a tobamovirus with a similar mode of transmission as PepMV that has the potential to severely affect tomato production worldwide [130].

## 5. Mechanism(s) of Mild Strain Cross-Protection

A common explanation of cross-protection is RNA silencing. During RNA silencing, double stranded RNA (dsRNA) produced during viral replication is recognized by Dicer-like (DCL) enzymes and cleaved into small fragments, 21–26 nucleotides long [131,132]. These small nucleotide fragments are called “small-interfering RNAs” (siRNAs) and are able to increase the antiviral RNA silencing response of the host plant by forming an RNA-induced silencing complex (RISC) together with Argonaute proteins. RISC has RNase III activity and degrades RNAs that share sequence identity with the siRNA that was used to form the RISC [131]. By dispersing siRNAs throughout the plant, systemic silencing occurs whereby viral silencing is induced in parts of the plant that had no previous viral contact. In this way RNAs that are similar in sequence to the RNA that first triggered the silencing response are broken down in all parts of the plant during systemic silencing. The sequence fidelity of this model may explain why cross-protection only works between closely related viruses and why genetically divergent viruses are able to evade recognition by RISC. In addition, the Argonaute proteins have also been implicated in translational repression of viral RNAs, induced by a combination of host resistance genes and viral elicitors [133,134]; whether this or degradation underlies MCSP requires further research.

RNA silencing by itself is not able to explain the mechanism of cross-protection fully. In a study done by Ziebell, Payne, Berry, Walsh, and Carr [135] a 2b silencing suppressor protein deletion mutant of cucumber mosaic virus (CMV) was unable to induce the strong systemic silencing signal expected of a virus lacking an RNA silencing suppressor, although it did provide limited protection against its parental strain (Fny-CMV) in *Nicotiana tabacum* and *Nicotiana benthamiana*. Interestingly, both strains appeared to be highly localized on a cellular level, with the two viruses occupying different cells, a phenomenon that has also been observed for other virus systems [136]. The authors suggested that cross-protection occurs either via very highly localized RNA silencing or via the competition between protective and challenge strains for host cells and resources. Furthermore, the observed protection against the parental Fny-CMV strain also occurred in silencing-deficient *Arabidopsis thaliana* plants, casting more doubt that RNA silencing was involved with at least this virus system [137]. Another study by Takeshita, Shigemune, Kikuhara, Furuya, and Takanami [138] supported the notion that RNA silencing by itself is unable to explain the phenomenon of cross-protection fully. The authors showed that different strains of CMV can spatially exclude each other from tissues in cowpea plants.

An alternative model explaining cross-protection without RNA silencing involves the protective strain preventing the challenge strain from uncoating upon entering the plant cell. The coat protein from the protective mild strain may recoat the challenge strain and prevent its replication [139], a proposal refuted based on effective protective strains of tobacco mosaic virus that produce no coat protein [140], or prevent uncoating by an undefined mechanism.

Superinfection-exclusion was proposed as the mechanism by which CTV cross-protection occurs [23], where it was observed that isolates are unable to superinfect a plant that has been pre-inoculated with an isolate of the same genotype. Homology between the pre- and superinfecting isolates is key, suggesting a genetic component is involved [141]. However, further research into this phenomenon found that the relative fitness between the protective and challenge isolates can determine whether superinfection can occur, thus a “weak” protective isolate can be superinfected by a more “adapted” challenge isolate [142]. Furthermore, the exclusion phenomenon was not uniform throughout the plant, and superinfection occurred between even near-identical isolates in the roots and lower stem of a plant [142].

In citrus at least, extant protective isolates such as GSMS-12 and Pera/IAC are populations, mixtures of different genotypes [143,144], suggesting another potential mechanism, the interaction of different isolates to alter fitness of one or more components of the population. It has been demonstrated that virus–virus interaction can alter isolate fitness, titer, and tropism through complementation or antagonism [33,145,146], and further research has found that these interactions can also be disrupted, and/or target suppressed genotypes [146], (*Harper* unpublished). It may be that a protective mixture is able to incorporate a pathogenic challenge isolate into the population, and reduce its fitness such that it cannot accumulate or express to a level necessary for pathogenesis.

The cross-protection phenomenon may not have a simple or single explanation. While some studied MSCP strains protect against infection with challenge isolates by triggering RNA silencing, others do not [10]. In some cases a specific viral protein is required for cross-protection, as is the case in the CTV pathosystem where p33 is required for cross-protection [147]. These experiments demonstrate that although cross-protection has been known to be an effective strategy in protecting plants from severe symptom development for almost a century, the exact mechanism of cross-protection, and therefore its efficacy constraints, remains obscure.

## 6. The Mild Strain Cross-Protection Era

Cross-protection as a disease management strategy can be very successful, as proven by the management of CTV or PepMV. Eradication of CTV by removing infected trees has proven unsuccessful, as the virus spreads quicker than infected trees can be identified and subsequently removed [10]. Using cultural practices and maintaining the populations of insect vectors at low levels to reduce pathogenic virus spread may be sufficient, however, these methods are often inadequate in lowering disease incidence.

For traditional breeding or cis-genics these are seldom the required genetic sources of resistance available within a cultivar, e.g., zucchini squash does not harbor sources of resistance to papaya ringspot virus type W (PRSV-W) within its genome [76]. While other management strategies, like genetically engineered resistance may be more effective in preventing disease incidence and spread, cross-protection offers a non-genetically modified (GMO) option for disease control that is more likely to be accepted by the public.

Societal and environmental pressures demand high quality and resilient cropping plants and produce with the use of low or no synthetic chemical inputs. Pathogenic viruses result in significant losses in crop production and crop quality but these are increasingly managed by MSCP which is both environmentally safe and socially friendly. Since MSCP is dependent on host and virus strains, an effort in finding local mild strains has to be made; any one mild strain cannot be used for the protection of cultivars grown in different locations. New technologies are converging to support the discovery and mechanism(s) of MSCP strains thereby accelerating the popularity of their use. However, without bioinformatic predictions of host range or pathogenicity the major barrier to the use of MSCP strains is in demonstrating their efficacy. Once efficacy is secured, MSCP proffers effective and sustainable protection against pathogenic viruses of crop plants across a range of production environments. The analysis of the societal and environmental benefits from the application of MSCP technology may provide impetus to further accelerate its use and, perhaps, assist in clarifying its underlying mechanism(s).

## Figures and Tables

**Table 1 viruses-11-00670-t001:** Protective virus isolates and their respective challenging isolates tested for their mild strain cross-protection capabilities. Host plants used to study the interaction of the protecting and challenging isolates, as well as the test site (Lab* and/or Field) are given. Viruses are listed according to the genus of the protecting virus.

Protecting Virus	Challenging Virus	Host Plant	Test Site	Year of Publication	Reference
**Alfamovirus**					
Alfalfa mosaic virus (AMV), strain 425	AMV, T6 strain	*Phaseolus vulgaris*	Lab	1995	[39]
**Badnavirus**					
Cocoa swollen shoot virus (CSSV), N1 and SS365B isolates	CSSV, 1A isolate	*Theobroma cacao* various cultivars	Field	2016	[40]

**Carmovirus**					
Turnip crinkle virus, TCVΔCP	TCV, T1d1	*Arabidopsis thaliana*	Lab	2014	[41]
**Caulimovirus**					
Cauliflower mosaic virus (CaMV), UN130 isolate	CaMV, Cabb S strain	*Brassica rapa*	Lab	1989	[42]
**Closterovirus**					
Citrus tristeza virus (CTV), T30 isolates	Citrus vein enation virus	*Citrus aurantifolia* cv. Mexican lime	Lab	2004	[43]
CTV, CS-1 isolate	CTV, Rolândia strain	*Citrus sinensis*	Field	2013	[44]
CTV, GFMS-12, GFMS-35, and LMS-6 strains	CTV severe strains	*C. paradisi* Macfad., *C. aurantifolia*, *C. sinensis*, *C. reticulata*	Field	2010, 2015	[45,46]

CTV	CTV	*Citrus sinensis* various cultivars, *Citrus × paradisi* various cultivars	Field	2010	[47]
**Cucumovirus**					
Cucumber mosaic virus (CMV), S strain	CMV, P strain	*Solanum lycopersicum* cv Rutgers, *Nicotiana tabacum* cvs Xanthi-nc, Turkish, *Cucurbita pepo*	Field	1985	[48]
CMV-S with S-CARNA 5	CMV severe strains	*Solanum lycopersicum,* various cultivars, *Cucurbita melo* cv. Janus des Canaries, *Capsicum annuum* cv. California Wonder	Field	1991, 1998	[49,50]
CMV, KO2 strain	Severe strain CMV-876	*Solanum lycopersicum*	Lab	1993	[51]
CMV, CM95 strain	CMV severe strains	*Cucumis sativus* cv. Sagamihanjiro	Lab	1997	[52]
Tomato aspermy virus (TAV) mild strain	TAV severe strains	*Solanum lycopersicum,* various cultivars	Lab	1986	[53]
**Furovirus**					
Beet soilborne mosaic virus	Beet necrotic yellow vein virus	*Beta vulgaris*	Lab	1999	[54]
**Carlavirus**					
Potato virus M (PVM), I-38 isolate	PVM, Uran isolate	*Datura metel*	Lab	2016	[55]
**Geminivirus**					
African cassava mosaic virus (ACMV) - Uganda	Virulent ACMV strains	*Manihot esculenta*	Field	2004	[56]
African cassava mosaic virus (ACMV)	East African cassava mosaic Cameroon virus (EACMVC)	*Nicotiana benthamiana*	Lab	2012	[57]
**Ilarvirus**					
Apple mosaic virus (ApMV), mild strain “M”	ApMV, severe strain “A”	*Malus sp*., various cultivars	Field	1964	[58]
**Luteovirus**					
Barley yellow dwarf virus (BYDV), mild isolates	BYDV severe strains	*Avena sativa* var. Clintland 64	Lab	1965, 1991	[59,60]
Potato leaf roll virus (PLRV), mild strain	PLRV severe strain	*Solanum tuberosum* ‘Katahdin’ *Physalis floridana*	Lab	1955	[61]
**Macluravirus**					
Chinese yam necrotic mosaic virus (CYNMV), KM3 strain	CYNMV wild-type	*Dioscorea opposita*	Field	2015	[62]
**Nepovirus**					
Grapevine fanleaf virus (GFLV), GHu strain	GFLV severe strain	*Vits vinifera* cv. Gewurztraminer clone 643 grafted onto rootstock Kober 5BB clone 259., *V. vinifera* various cultivars	Field	2008, 1993	[63,64]
Arabis mosaic virus (ArMV), Ta strain	GFLV severe strain	*Vits vinifera* cv. Gewurztraminer clone 643 grafted onto rootstock Kober 5BB clone 259., *V. vinifera,* various cultivars	Field	2008, 1993	[63,64]
Tomato ringspot virus (ToRSV), Chickadee isolate	ToRSV, PYBM isolate	*Nicotiana benthamiana*	Lab	1988	[65]
**Potexvirus**					
Pepino mosaic virus (PepMV), LP,EU, and CH2 strains	Severe PepMV, CH2 isolate	*Solanum esculentum* cv. Tricia	Lab	2010	[66]
PepMV, VX1, and VC1 mixture (LP and CH2 genotype)	Severe PepMV isolates of EU and CH2 genotype	*Solanum esculentum* various cultivars	Lab	2017	[67]
PepMV, Sp13, and PS5	Severe PepMV isolates of EU and CH2 genotype	*Solanum esculentum* various cultivars	Field	2018	[68]
PepMV, KD strain (engineered)	PepMV, wild-type	*Nicotiana benthamiana*, *Solanum esculentum* various cultivars	Lab	2015	[69]
Potato virus X (PVX), mild strain	PVX severe strain	*Nicotiana* *tabacum*	Lab	1933	[36]
PVX E1001A, E46A	Wild type PVX severe strain	*Nicotiana benthamiana*	Lab	2019	[70]
**Potyvirus**					
Bean yellow mosaic virus (BYMV), M11 isolate	BYMV severe strains and *Clover yellow vein virus*	*Vicia faba*	Lab	2009	[71]
Cowpea aphid-borne mosaic virus (CABMV), cowpea and passionfruit isolates	various combinations of severe isolates	*Canavalia ensiformis*, *Passiflora edulis*	Lab	2017	[72]
Maize dwarf mosaic virus (MDMV)	Bermuda grass Southern mosaic virus (BgSMV)	*Sorghum* (unknown species)	Lab	2012	[73]
Papaya ringspot virus (PRSV), PRSV HA 5-1 and 6-1 mutated strains	PRSV-W severe strains PRSV-W-C, PRSV-W-B, PRSV-W-P	*Carica papaya*	Field	1998	[74]
Papaya ringspot virus (PRSV), PRSV-W-SD mutated strains C57A, K125D, G317K, P328A	Wild type PRSV-W-gfp	*Cucumis melo*	Lab	2019	[75]
PRSV-W mild strains PRSV-W-1 and PRSV-W-2	PRSV-W severe strains PRSV-W-C, PRSV-W-B, PRSV-W-P	*Cucurbita pepo* cv. Caserta and cv. Clarinda	Field	1998	[76]
Pepper severe mosaic virus (PeSMV), M-1 strain	PeSMV virulent strains	*Capsicum* cv. XPH 833 and cv. NUN 3364	Lab	1988	[77]
Plum pox virus (PPV) mild strains	PPV severe strains	*Prunus persica* GF305	Lab	2008	[78]
Potato virus Y (PVY) mild strains M-MY10 and N-NA10	PVY severe isolate NTND6	*Nicotiana tabacum* cv. Xanthi	Lab	2013	[79]
PVY isolates Li, FrKV2, Wy, Wi, Cou8/03, 47/96	various combination of PVY isolates	*Nicotiana tabacum* cv. Samsun, *S. tuberosum* cv Irga and cv. Satina	Lab	2018	[80]
Potato virus A (PVA) tobacco strain	PVA potato strains	*Nicotiana tabacum* cv. Samsun nn	Lab	2002	[81]
Soybean mosaic virus (SMV), Aa15-M2 strain	SMV severe strains	*Glycine max* cv. Shin Tambaguro	Lab and field	1993	[82]
SMV, mosaic strain	SMV mosaic strain	*Glycine max,* various cultivars	Lab	2008	[83]
Turnip mosaic virus, GK strain	YC5 strain	*Arabidopsis thaliana*	Lab	2014	[84]
Vanilla necrosis potyvirus (VNV), mild strains	VNV severe strains	*Vanilla fragrans*	Lab	1999	[85]
Watermelon mosaic virus (WMV), EM, CL isolates; W1-9 isolates	WMV, EM, and CL isolates and WMV severe strains	*Cucurbita maxima* cv. Hokou Aokawa Kuri, *C. moshata*, various cultivars, *C. maxima x C. moshata,* various cultivars, *C. pepo*, *Cucumis melo* cv. various, *Citrullus lunatus* cv. Kyokuto, *Lagenaria siceraria* cv. Shimotsukeshiro	Lab	1992, 2011	[86,87]

Zucchini yellow mosaic virus (ZYMV), WK isolate	ZYMV severe strains	*Curcubita pepo* cvs. Elite, Oriental sweet melon, *Cucumis sativus* cv. Marketer	Lab and field	1991	[19]
**Rymovirus**					
Wheat streak mosaic virus (WSMV), Type and Sidney 81 strains	WSMV, Type and Sidney 81 strains	*Triticum aestivum* cv. Centurk	Lab	2001	[88]
**Tobamovirus**					
Cucumber green mottle mosaic virus (CGMMV), VIROG-43Ms strain	CGMMV, MC-1 and MC-2 isolates	*Cucumis sativus* var. Kurazh F1	Lab	2016	[89]
CGMMV Pk-47 and Pk-81	Wild type CGMMV	*Cucumis sativus*	Lab	2010	[90]
Hibiscus latent Singapore virus (HLSV)	Tobacco mosaic virus-U1	*Nicotiana benthamiana*	Lab	2013	[91]
Pepper mild mottle virus (PMMoV), isolates L3-163	Wild type PMMoV isolate	*Capsicum annuum,* various cultivars	Lab	2013	[92]
Tobacco mosaic virus (TMV), 43A	Wild type TMV	*Nicotiana benthamiana*	Lab	2019	[93]
TMV, MII-16 strain	TMV, type O isolate	*Solanum lycopersicum* cv. Potentate	Lab	1977	[94]
TMV mild strain	TMV	*Capscicum annuum* various cultivars	Lab	1984	[95]
Crucifer TMV, CgYD strain (engineered)	Crucifer TMV, Cg strain	*Arabidopsis thaliana* ecotype Col-0 and sde1 (ecotype C24)	Lab	2003	[96]
Satellite TMV, type 5 and type 6	Satellite TMV, type 5 and type 6	*Nicotiana**tabacum* cv. Xanthi nn	Lab	1994	[97]
Tomato mosaic virus (ToMV), L_11_A-Fukushima strain	ToMV severe strains	*Solanum lycopersicum* cv. Momotaro	Field	2002	[98]
**Tospovirus**					
Tomato spotted wilt virus (TSWV), R27G mild strain	TSWV, BL isolate	*Datura stramonium* L.	Lab	1992	[99]

* Lab = Research laboratory or glasshouse.

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
