# Peer review of "A New Era for Mild Strain Cross-Protection"

_viruses, 2019, doi:10.3390/v11070670_

Round 1

Reviewer 1 Report

This manuscript makes the case that with societal pressure rising against the use of chemicals or GMOs, cross-protection with mild virus strains will increasingly become the method of choice.  The manuscript is interesting, well-written and generally offers an accurate account of the potential benefits and limitations of the technique.  I only have a few minor suggestions for improvement.

Cross-protection has been known and used for a very long time.  Therefore the title "The mild strain cross-protection era" is perhaps a bit misleading.  I would suggest a more precise title perhaps "A new era for mild strain cross-protection" or "A revival of mild strain cross-protection approaches".  Similarly, to make the point that cross-protection is not really a new concept, I wonder if it would not be a good idea to add dates in the table.  Some of the examples chosen are from the fifties, sixties etc... Adding date would give the reader a more accurate impression that although cross-protection has been discovered, tested and used for a long time,  there is a recent renewed interest.  Finally, on line 118, I would suggest to add a few key references in support of "the cross-protection phenomenon was observed a century ago".

The authors touch on the concept that mild strains need to be tested for their stability and for their virulence in various hosts (lines 58-61).  However, perhaps this needs to be expanded a little either here or later in the text, as this is a critical point.  Are there documented examples of studies confirming that mild strains do not usually evolve over expanded periods of use to give rise to more pathogenic isolates?  Are there examples of mild strains that escaped to alternate crops in a more virulent forms?  Are those risks within the acceptable range (i.e. risk vs benefit)?  

line 23: change to "discovery and mechanism(s) of action of MSCP strains"

line 138 delete "of it"

line 148 change "resources" to "sources"

line 178-179 there is increasing evidence that antiviral RNA silencing can also operate by repressing the translation of viral genomes.  This should be mentioned.

Author Response

Reviewer 1:

This manuscript makes the case that with societal pressure rising against the use of chemicals or GMOs, cross-protection with mild virus strains will increasingly become the method of choice.  The manuscript is interesting, well-written and generally offers an accurate account of the potential benefits and limitations of the technique.  I only have a few minor suggestions for improvement.

Cross-protection has been known and used for a very long time.  Therefore the title "The mild strain cross-protection era" is perhaps a bit misleading.  I would suggest a more precise title perhaps "A new era for mild strain cross-protection" or "A revival of mild strain cross-protection approaches".

Title changed to: A new era for mild strain cross-protection

Similarly, to make the point that cross-protection is not really a new concept, I wonder if it would not be a good idea to add dates in the table.  Some of the examples chosen are from the fifties, sixties etc... Adding date would give the reader a more accurate impression that although cross-protection has been discovered, tested and used for a long time,  there is a recent renewed interest. 

Added a column in the table with the year of publication.

Finally, on line 118, I would suggest to add a few key references in support of "the cross-protection phenomenon was observed a century ago".

“Historic” references from McKinney, Salaman added and also the review by Ziebell & Carr cross-referenced

The authors touch on the concept that mild strains need to be tested for their stability and for their virulence in various hosts (lines 58-61).  However, perhaps this needs to be expanded a little either here or later in the text, as this is a critical point.  Are there documented examples of studies confirming that mild strains do not usually evolve over expanded periods of use to give rise to more pathogenic isolates?  Are there examples of mild strains that escaped to alternate crops in a more virulent forms?  Are those risks within the acceptable range (i.e. risk vs benefit)? 

A section discussing these points has been added (lines 71ff)

line 23: change to "discovery and mechanism(s) of action of MSCP strains"

Changed

line 138 delete "of it"

Changed

line 148 change "resources" to "sources"

Changed

line 178-179 there is increasing evidence that antiviral RNA silencing can also operate by repressing the translation of viral genomes.  This should be mentioned.

This has been acknowledged in the text (now lines 196ff).

Reviewer 2 Report

The article is very interesting, provides a new  and important contribution in the field and explore the possibility to apply MSCP to control viruses.  

The examples given in the work are appropriate and well described and interpretated.

The work is scientifically relevant, very well revised, well organized and clearly written.

 Also, the literature is appropriate to the topic of the paper, complete and updated. 

Author Response

The article is very interesting, provides a new  and important contribution in the field and explore the possibility to apply MSCP to control viruses.  

The examples given in the work are appropriate and well described and interpretated.

The work is scientifically relevant, very well revised, well organized and clearly written.

 Also, the literature is appropriate to the topic of the paper, complete and updated. 

Thank you very much for your kind evaluation!